# Renal function in a cohort of HIV-infected patients initiating antiretroviral therapy in an outpatient setting in Ethiopia

Temesgen Fiseha[1]*, Angesom Gebreweld[2]

1 Department of Clinical Laboratory Science, College of Medicine and Health Sciences, Wollo University, Dessie, Ethiopia, 2 Department of Medical Laboratory Science, College of Health Sciences, Mekelle University, Mekelle, Ethiopia

* temafiseha@gmail.com

## Abstract

### Aim

To evaluate the prevalence and associated factors of abnormal renal function among Ethiopian HIV-infected patients at baseline prior to initiation of antiretroviral therapy (ART) and during follow-up.

### Methods

We conducted a retrospective observational cohort study of HIV infected patients who initiated ART at the outpatient ART clinic of Mehal Meda Hospital of North Shewa, Ethiopia from January 2012 to August 2018. Demographic and clinical data were abstracted from the medical records of patients. Renal function was assessed by estimated glomerular filtration rate (eGFR) calculated using the Modification of Diet in Renal Disease (MDRD) equation. Univariate and multivariate analysis were conducted to determine the factors associated with abnormal renal function at baseline and during follow-up.

### Results

Among 353 patients, 70 (19.8%) had baseline eGFR <60 ml/min/1.73m$^2$ and 102 (28.9%) had eGFR = 60–89.9 ml/min/1.73m$^2$. Factors associated with baseline renal impairment (eGFR <60 ml/min/1.73m$^2$) included female sex (AOR = 3.52, CI 1.75–7.09), CD4 count < 200 cells/mm$^3$ (AOR = 2.75, CI 1.40–5.42), BMI < 25 Kg/m$^2$ (AOR = 3.04, CI 1.15–8.92), low hemoglobin (AOR = 2.19, CI 1.16–4.09) and high total cholesterol (AOR = 3.15, CI 1.68–5.92). After a median of 3.0 years of ART, the mean eGFR declined from 112.9 ± 81.2 ml/min/1.73m$^2$ at baseline to 93.9 ± 60.6 ml/min/1.73m$^2$ ($P$ < 0.001). The prevalence of renal impairment increased from 19.8% at baseline to 22.1% during follow-up. Of 181 patients with baseline normal renal function, 49.7% experienced some degree of renal impairment. Older age (AOR = 3.85, 95% CI 2.03–7.31), female sex (AOR = 4.18, 95% CI 2.08–8.40), low baseline CD4 (AOR = 2.41, 95% CI 1.24–4.69), low current CD4 count (AOR = 2.32, 95% CI 1.15–4.68), high BMI (AOR = 2.91, 95% CI 1.49–5.71), and low

**Data Availability Statement:** All relevant data are within the paper and its Supporting Information files.

**Funding:** The authors received no specific funding for this work.

**Competing interests:** The authors have declared
that no competing interests exist.

**Abbreviations:** AOR, Adjusted odds ratio; ART,
antiretroviral therapy; BMI, body mass index; CI,
Confidence interval; eGFR, estimated glomerular
filtration rate; MDRD, Modification of Diet in Renal
Disease; OR, odds ratio.

hemoglobin (AOR = 3.38, 95% CI 2.00–7.46) were the factors associated with renal
impairment during follow-up.

## Conclusion

Impaired renal function was common in HIV-infected patients initiating ART in an outpatient
setting in Ethiopia, and there appears to be a high prevalence of renal impairment after a
median ART follow-up of 3 years. There is a need for assessment of renal function at base-
line before ART initiation and regular monitoring of renal function for patients with HIV during
follow-up.

## Introduction

Impaired renal function is a common complication of HIV infection in both the pre-antiretro-
viral therapy (ART) and ART periods, and is associated with increased risk of HIV disease pro-
gression, AIDS- and non-AIDS-related events, cardiovascular disease and mortality [1–5]. In
patients with HIV, impaired renal function has been shown to be a risk factor for drug-related
adverse events, acute renal failure, hospital admissions and progression to end-stage renal dis-
ease requiring renal replacement therapies in the form of dialysis or transplantation, which
may impact quality of life and survival [5–8]. As a consequence, assessment of renal function is
recommended for all HIV-infected individuals at baseline prior to initiation of ART and dur-
ing follow-up [9].

Renal impairment during the pre-ART period was largely a result of HIV-associated renal
disease, which is a leading cause of chronic kidney disease and end-stage renal disease, espe-
cially in persons of African descent and is caused by direct injury to the kidneys by the HIV
[10,11]. Although the incidence of HIV-associated renal impairment has decreased in the
recent ART period, the proportion of renal impairment has increased due to increasing bur-
den of comorbid renal impairment risk factors such as diabetes and hypertension in this aging
population [12–14]. The increase in the incidence of renal impairment could also be related to
the long-term exposure to potentially nephrotoxic antiretrovirals, such as tenofovir, ritonavir-
boosted atazanavir, and ritonavir-boosted lopinavir, or to the effects of HIV infection itself,
pre-existing renal impairment and of co-infections [15–18].

Renal impairment was seen in up to 24% of patients at baseline prior to ART initiation [19]
and there are numerous studies demonstrating renal function improvement after initiation of
ART [20,21]. Others have reported no significant improvement and/or progressive loss of
renal function during follow-up [22–24]. Given the high burden of HIV and the introduction
of the "test-and treat" strategy in sub-Saharan Africa, an understanding of the prevalence and
associated factors of renal impairment before and during ART follow-up is imperative to
guide strategies in the management and clinical care of this population. Despite the introduc-
tion of the 'test and treat' program in Ethiopia, little is known about the status of renal function
in HIV-infected individuals at the time of initiation of ART and during ART in this resource
constrained settings. Renal function test prior to and during ART is not routinely performed.
The aim of our study was therefore to evaluate the prevalence and associated risk factors of
abnormal renal function at baseline and during ART follow-up and to describe the impact of
ART on renal function in a cohort of HIV-infected patients initiating ART in an outpatient
setting in Ethiopia.

## Methods

### Study design and population

This was a retrospective observational cohort study of HIV infected patients who initiated ART at the outpatient ART clinic of Mehal Meda Hospital of North Shewa, Ethiopia from January 2012 to August 2018. A total of 353 HIV-infected patients on ART follow-up were included in the study. The inclusion criteria were 18 years and older, receipt of ART for 6 months or more, and measured serum creatinine results at the baseline prior to initiation of ART and during follow-up (after the initiation of ART). Patients with missing data for essential variables, pregnant, and end-stage renal disease on dialysis patients were excluded from the study analysis. The study protocol was approved by the Institutional Review Board of College of Medicine and Health Sciences, Wollo University (# 133/13/12). Written informed consent was not given by participants for their clinical records to be used in this study, but patient's identifiers were removed and only code numbers were used throughout the study. The database for this manuscript is available as an additional supporting information (S1 Dataset).

### Data collection

Data were abstracted from the medical records of the patients. The data obtained included demographics (e.g., age, sex, residence, education, height, and body weight), history of diabetes and/or hypertension and clinical information (e.g., CD4 cell count, World Health Organization (WHO) HIV/AIDS clinical stage, hemoglobin test results, serum total cholesterol and serum creatinine, duration of ART use, and types of ART regimens). Serum creatinine was measured and recorded at baseline prior to ART initiation, at 3 and 6 months after initiating therapy, and annually thereafter. After initiation of therapy, if a patient had multiple documented serum creatinine results, only the most recent result was utilized for the analysis. Renal function was assessed by estimated glomerular filtration rate (eGFR) calculated using the Modification of Diet in Renal Disease (MDRD) equation [25], and categorized into 3 categories (eGFR $\geq$90, 60–89.9 and < 60 ml/min/1.73 m$^2$) according to the National Kidney Foundation criteria [26]. Patients were defined as having renal impairment at baseline and during follow up if eGFR values were < 60 ml/min/1.73 m$^2$.

### Statistical analysis

Patient data were entered into an "EpiData version 3.1" and analyzed using SPSS version 20 software (SPSS Inc., Chicago, IL, USA). Categorical variables were summarized by frequencies and percentages, and continuous variables were summarized by mean ± standard deviation (SD). Paired t-test was used to evaluate mean eGFR and CD4 T cell count difference before and after ART initiation. McNemar's test was used to compare the proportion of patients with renal impairment before and after ART initiation. Univariate analyses to assess factors associated with renal impairment at baseline and during follow-up were tested using Chi-square test and student's t-test, where appropriate. Variables that were found to be significant in univariate analysis ($P < 0.25$) were included in the multivariate backwards stepwise logistic regression model to identify factors independently associated with renal impairment. $P$-values < 0.05 were considered significant.

## Results

Six hundred and forty-eight adults $\geq$ 18 years old were followed in this outpatient ART clinic for at least six months. A total of 295 were excluded: 143 did not have recorded baseline creatinine result before ART initiation, 40 did not have creatinine results after initiating ART at the

most recent visit, and 112 patients had missing data or lost to follow up (Fig 1). The excluded patients were similar to those who were included in this analysis in terms of age (mean 37.3 ± 10.0 years) and gender (60.0% female) distributions. The mean baseline CD4 count was 398.6 ± 217 cells/mm$^3$ in the excluded HIV-infected patients and 50 (16.9%) were in WHO clinical stage 3/4 (See S1 Table, which shows baseline characteristics of patients included and excluded from the analysis).

A total of 353 HIV-infected patients on ART with serum creatinine data available at the baseline and during follow-up were included in the study. The baseline demographic and clinical characteristics of the patients are summarized in Table 1. Mean (± SD) age was 37.5 ± 9.7 years and 58.9% were females. Mean BMI was 19.6 ± 4.1 kg/m$^2$; and only 16.7% were overweight/obese (BMI ≥ 25 kg/m$^2$). The mean baseline CD4 count was 406.5 ± 274 cells/mm$^3$, with 25.5% of patients presenting with a CD4 count of <200 cells/mm$^3$. An advanced HIV disease stage (WHO stage 3/4) was present in 18.4% of patients at baseline. The mean hemoglobin, total cholesterol and serum creatinine were 13.6 ± 2.9 g/dl, 177.4 ± 48.5 mg/dl and 1.06 ± 0.48 mg/dl respectively. Twenty-seven (7.6%) patients had a history of diabetes or hypertension at baseline.

## Renal function at baseline

The mean baseline eGFR of the study patients was 112.9 ± 81.2 ml/min/1.73m$^2$. A total of 70 (19.8%) patients had baseline eGFR <60 ml/min/1.73m$^2$, 102 (28.9%) had eGFR = 60–89.9 ml/min/1.73m$^2$, and 181 (51.3%) had eGFR ≥ 90 ml/min/1.73m$^2$. In univariate analysis, age over 50 years (OR = 3.29, 95% CI 1.57–6.90) and female sex (OR = 2.82, CI 1.54–5.18) were associated with renal impairment (eGFR < 60 ml/min/1.73 m$^2$) at baseline before ART initiation. Low CD4 count, low BMI, low hemoglobin, and high total cholesterol were also associated with renal impairment. No association was found with residence, education, WHO clinical disease stage and comorbid diabetes or hypertension (Table 2). Multivariable logistic regression analysis identified female sex (AOR = 3.52, CI 1.75–7.09; $P$ < 0.001), CD4 count < 200 cells/mm$^3$ (AOR = 2.75, CI 1.40–5.42; $P$ = 0.004), BMI < 25 Kg/m$^2$ (AOR = 3.04, CI 1.15–8.92; $P$ = 0.030), low hemoglobin < 12 g/dl (AOR = 2.19, CI 1.16–4.09; $P$ = 0.015),

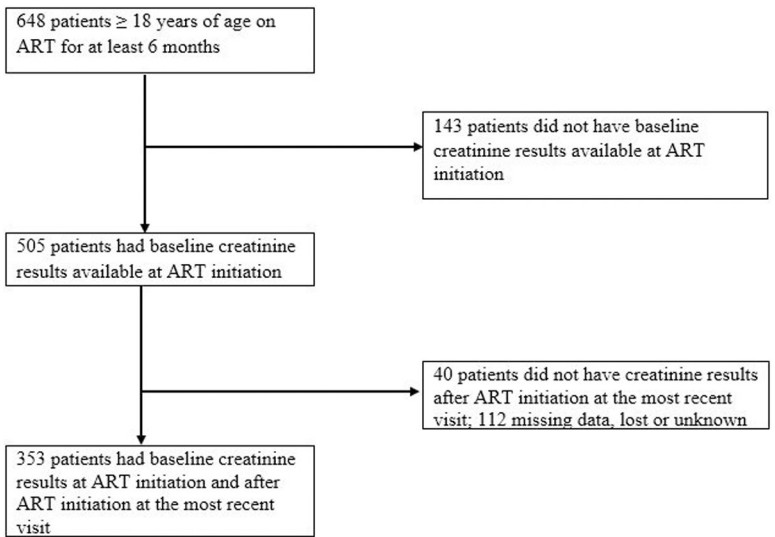

**Fig 1. Flow diagram showing the number of patients included in the analysis.**

**Table 1. Baseline characteristics of the study patients (n = 353).**

| Characteristics | | |
|---|---|---|
| Age (year), mean ± SD | | 37.5 ± 9.7 |
| Age group, n (%) | | |
| | 18–30 years | 103 (29.2) |
| | 31–50 years | 228 (64.6) |
| | > 50 years | 22 (6.2) |
| Sex, n (%) | | |
| | Male | 145 (41.1) |
| | Female | 208 (58.9) |
| Residence, n (%) | | |
| | Urban | 227 (64.3) |
| | Rural | 126 (35.7) |
| Education, n (%) | | |
| | < High school | 291 (82.4) |
| | ≥ High school | 62 (17.6) |
| WHO clinical stage, n (%) | | |
| | 1/2 | 288 (81.6) |
| | 3/4 | 65 (18.4) |
| CD4 count (Cells/mm$^3$), mean ± SD | | 406.5 ± 274 |
| CD4 category, n (%) | | |
| | < 200 Cells/mm$^3$ | 90 (25.5) |
| | 200–350 Cells/mm$^3$ | 85 (24.1) |
| | > 350 Cells/mm$^3$ | 178 (50.4) |
| Body mass index (kg/m$^2$), mean ± SD | | 19.6 ± 4.1 |
| Diabetes or hypertension, n (%) | | |
| | Yes | 27 (7.6) |
| | No | 326 (92.4) |
| Hemoglobin (g/dl), mean ± SD | | 13.6 ± 2.9 |
| Total cholesterol (mg/dl), mean ± SD | | 177.4 ± 48.5 |
| Serum Creatinine (mg/dl), mean ± SD | | 1.06 ± 0.48 |

and high total cholesterol ≥ 200 mg/dl (AOR = 3.15, CI 1.68–5.92; $P < 0.001$) as risk factors for baseline renal impairment.

## Renal function on ART follow up

The median duration on ART was 3.0 years (interquartile range [IQR], 1.5–5.4 years) and 221 (62.6%) patients were on non-TDF based ART regimens. The mean CD4 count was an average of 96.2 cells/mm$^3$ higher than baseline (502.8 ± 289 vs 406.5 ± 274 cells/mm$^3$, $P < 0.001$). The mean eGFR declined from 112.9 ± 81.2 ml/min/1.73 m$^2$ at baseline to 93.9 ± 60.6 ml/min/ 1.73m$^2$ after a median ART follow-up of 3 years ($P < 0.001$). The prevalence of renal impairment (eGFR <60 ml/min/1.73 m$^2$) increased from 19.8% at baseline to 22.1% after a median of 3 years on ART. The proportion of patients with eGFR = 60–89.9 ml/min/1.73m$^2$ increased by 10.2% after ART. Of 70 patients with eGFR <60 ml/min/1.73m$^2$ at baseline, 40% still had eGFR <60 ml/min/1.73m$^2$, 40% experienced eGFR = 60–90 ml/min/1.73 m$^2$ and 20.0% recovered to eGFR ≥90 ml/min/1.73 m$^2$ after a median of 3 years of ART. For the 102 patients who had eGFR 60–90 ml/min/1.73 m$^2$ at baseline, 27.5% experienced a decreased in eGFR to <60 ml/min/1.73 m$^2$ and 31.3% improved to normal eGFR >90 ml/min/1.73m$^2$

**Table 2. Factors associated with renal impairment (eGFR < 60 ml/min/1.73 m$^2$) at baseline.**

| Characteristic | eGFR < 60 ml/min/1.73m$^2$ | | Crude OR (95% CI) | P-value | Adjusted OR (95% CI) | P-value |
|---|---|---|---|---|---|---|
| | Yes (n = 70) | No (n = 283) | | | | |
| Age (years) | | | | | | |
| < 50 | 56 (17.6) | 263 (82.4) | 1 | 0.001 | 1 | 0.174 |
| ≥ 50 | 14 (41.2) | 20 (58.8) | 3.29 (1.57–6.90) | | 1.75 (0.71–4.30) | |
| Sex | | | | | | |
| Male | 16 (11.0) | 129 (89.0) | 1 | 0.001 | 1 | 0.001 |
| Female | 54 (26.0) | 154 (74.0) | 2.82 (1.54–5.18) | | 3.52 (1.75–7.09) | |
| Residence | | | | | | |
| Urban | 48 (21.1) | 179 (78.9) | 1.27 (0.72–2.23) | 0.405 | | |
| Rural | 22 (17.5) | 104 (82.5) | 1 | | | |
| Education | | | | | | |
| < High school | 63 (21.6) | 228 (78.4) | 2.17 (0.94–5.00) | 0.063 | 1.81 (0.72–4.57) | 0.208 |
| ≥ High school | 7 (11.3) | 55 (88.7) | 1 | | | |
| CD4 count (cells/mm$^3$) | | | | | | |
| < 200 | 28 (31.4) | 62 (68.9) | 2.53 (1.38–4.63) | 0.003 | 2.75 (1.40–5.42) | 0.004 |
| 200–350 | 15 (17.6) | 70 (82.4) | 1.20 (0.60–2.39) | 0.608 | 1.36 (0.64–2.89) | 0.432 |
| > 350 | 27 (15.2) | 151 (84.8) | 1 | | 1 | |
| WHO Clinical Stage | | | | | | |
| 1/2 | 56 (19.4) | 232 (80.6) | 1 | 0.702 | | |
| 3/4 | 14 (21.5) | 51 (78.5) | 0.68 (0.46–1.70) | | | |
| Body mass index (Kg/m$^2$) | | | | | | |
| < 25 | 65 (22.1) | 229 (77.9) | 3.07 (1.18–7.98) | 0.017 | 3.04 (1.15–8.92) | 0.030 |
| ≥ 25 | 5 (8.5) | 54 (91.5) | 1 | | 1 | |
| Hemoglobin (g/dl) | | | | | | |
| < 12 | 27 (34.2) | 52 (65.8) | 2.79 (1.58–4.92) | < 0.001 | 2.19 (1.16–4.09) | 0.015 |
| ≥ 12 | 43 (15.7) | 231 (84.3) | 1 | | 1 | |
| Diabetes or hypertension | | | | | | |
| Yes | 9 (33.3) | 18 (66.7) | 2.17 (0.93–5.07) | 0.067 | 1.88 (0.76–4.67) | 0.226 |
| No | 61 (18.7) | 265 (81.3) | 1 | | 1 | |
| Total cholesterol (mg/dl) | | | | | | |
| < 200 | 43 (15.6) | 233 (84.4) | 1 | < 0.001 | 1 | <0.001 |
| ≥ 200 | 27 (35.1) | 50 (64.9) | 2.93 (1.66–5.17) | | 3.15 (1.68–5.92) | |

after a median of 3 years on ART. Among 181 patients with baseline normal eGFR ≥90 ml/min/1.73 m$^2$, 12.1% experienced a decrease in eGFR to <60 ml/min/1.73m$^2$ and 37.6% experienced eGFR = 60–89.9 ml/min/1.73m$^2$ after a median treatment duration of 3 years (Fig 2).

The univariate analysis demonstrated that older age (OR = 3.95, 95% CI 2.30–6.80), female sex (OR = 3.77, 95% CI 2.04–6.94), low baseline CD4 (OR = 2.42, 95% CI 1.41–4.13), current CD4 count (OR = 3.11, 95% CI 1.75–5.52), duration on ART (OR = 1.74, 95% CI 1.04–2.90), high BMI (OR = 3.38, 95% CI 1.92–5.95), low hemoglobin < 12 g/dl (OR = 3.67, 95% CI 2.09–6.47), and baseline renal impairment (OR = 3.11, 95% CI 1.76–5.48) were associated with renal impairment during ART follow-up (Table 3). We did not find any association between renal impairment and residence, education, WHO clinical disease stage, or treatment with a TDF-based regimen. In the multivariable analysis, older age (AOR = 3.85, 95% CI 2.03–7.31; P < 0.001), female sex (AOR = 4.18, 95% CI 2.08–8.40; P < 0.001), low baseline CD4 (AOR = 2.41, 95% CI 1.24–4.69; P = 0.009), low current CD4 count (AOR = 2.32, 95% CI 1.15–4.68; P = 0.018), high BMI (AOR = 2.91, 95% CI 1.49–5.71; P = 0.002), and low

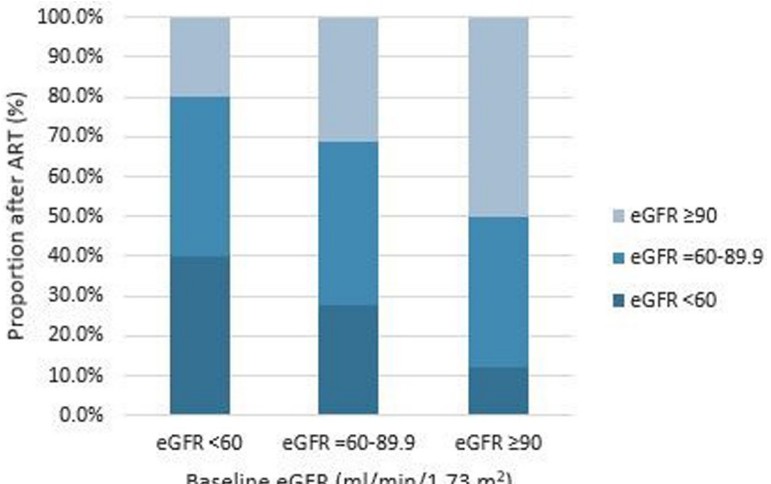

**Fig 2. Distribution of eGFR categories of HIV-infected patients on ART stratified by baseline eGFR.**

hemoglobin < 12 g/dl (AOR = 3.38, 95% CI 2.00–7.46; $P$ = 0.001) remained significantly associated with renal impairment.

## Discussion

In this study, we evaluated renal function among HIV-positive individuals at baseline prior to ART initiation and during follow-up. At baseline before ART initiation, 48.8% of the patients had eGFR < 90 ml/min/1.73 m$^2$. Of these 19.8% had eGFR < 60 ml/min/1.73 m$^2$ and 28.9% had eGFR 60–89.9 ml/min/1.73 m$^2$. Factors associated with baseline renal impairment (eGFR <60 ml/min/1.73m m$^2$) were female sex, lower CD4 count, lower BMI, lower hemoglobin and higher total cholesterol. Our findings are comparable to some of the studies in the region regarding the prevalence of renal impairment at baseline prior to ART initiation using the same definition with MDRD formula [21,27]. Other studies showed highly variable prevalence of renal impairment at baseline before initiation of ART. In a large cohorts of HIV-positive subjects, studies in Burkina Faso [28], China [29], Spain [30] and Nigeria [19] reported prevalence of renal impairment (eGFR < 60 ml/min/1.73 m$^2$) of 3.0%, 3.3%, 5% and 24%, respectively at baseline before ART initiation using the same MDRD formula for eGFR. The disparity seems to be explained mainly by the characteristics of studied patients or by the differences in the creatinine assays and calibration. It can thus be inferred that HIV as a disease state is associated with renal function impairment independent of the possible nephrotoxic effects of antiretroviral agents used in the treatment of persons with HIV. The basis of HIV-induced renal impairment has been recognized as direct cellular injury by the virus or by changes in the release of cytokines during HIV infection [19,31].

Our study documented risk factors for renal impairment (eGFR < 60 ml/min/1.73 m$^2$) in patients with HIV at the baseline prior to ART initiation that are similar to those identified by previous studies including female sex, low CD4 count, low BMI, and low hemoglobin [21,29,32–35]. In this study, older age at baseline was also significantly associated with increased risk of renal impairment before initiating ART in univariate analyses, but did not obtain statistical significance in multivariate analysis. Multiple previous studies in cohorts of HIV-positive subjects initiating ART reported older age as a strong risk factor of impaired renal function at treatment initiation [21,29,30,33–35]. Interestingly in our study we found higher total cholesterol associated with baseline renal impairment in patients with HIV. The

**Table 3. Factors associated with renal impairment (eGFR $<$ 60 ml/min/1.73 m$^2$) during ART follow-up.**

| Characteristic | eGFR $<$60 ml/min/1.73m$^2$ | | Crude OR (95% CI) | P-value | Adjusted OR (95% CI) | P-value |
|---|---|---|---|---|---|---|
| | Yes (n = 78) | No (n = 275) | | | | |
| Age (years) | | | | | | |
| $<$ 50 | 42 (15.7) | 226 (84.3) | 1 | $<$ 0.001 | 1 | $<$ 0.001 |
| $\geq$ 50 | 36 (42.4) | 49 (57.6) | 3.95 (2.30–6.80) | | 3.85 (2.03–7.31) | |
| Sex | | | | | | |
| Male | 15 (10.3) | 130 (89.7) | 1 | $<$ 0.001 | 1 | $<$ 0.001 |
| Female | 63 (30.3) | 145 (69.7) | 3.77 (2.04–6.94) | | 4.18 (2.08–8.40) | |
| Residence | | | | | | |
| Urban | 56 (24.7) | 171 (75.3) | 1.55 (0.89–2.68) | 0.118 | 1.06 (0.56–2.03) | 0.858 |
| Rural | 22 (17.5) | 104 (82.5) | 1 | | 1 | |
| Education | | | | | | |
| $<$ High school | 65 (22.3) | 226 (77.7) | 1.08 (0.55–2.12) | 0.814 | | |
| $\geq$ High school | 13 (21.0) | 49 (79.0) | 1 | | | |
| Baseline CD4 count (cells/mm$^3$) | | | | | | |
| $<$ 200 | 31 (34.4) | 59 (65.6) | 2.42 (1.41–4.13) | 0.001 | 2.41 (1.24–4.69) | 0.009 |
| $\geq$ 200 | 47 (17.9) | 216 (82.1) | 1 | | 1 | |
| Current CD4 count (cells/mm$^3$) | | | | | | |
| $<$ 200 | 27 (40.3) | 40 (59.7) | 3.11 (1.75–5.52) | $<$ 0.001 | 2.32 (1.15–4.68) | 0.018 |
| $\geq$ 200 | 51 (17.8) | 235 (82.2) | 1 | | 1 | |
| WHO Clinical Stage | | | | | | |
| 1/2 | 67 (22.1) | 236 (77.9) | 1.01 (0.49–2.07) | 0.986 | | |
| 3/4 | 11 (22.0) | 39 (78.0) | 1 | | | |
| Body mass index (Kg/m2) | | | | | | |
| $<$ 25 | 49 (17.3) | 234 (82.7) | 1 | $<$ 0.001 | 1 | 0.002 |
| $\geq$ 25 | 29 (41.4) | 41 (58.6) | 3.38 (1.92–5.95) | | 2.91 (1.49–5.71) | |
| Duration on ART (years) | | | | | | |
| $\leq$ 3 | 31 (17.4) | 147 (82.6) | 1 | 0.033 | 1 | 0.269 |
| $>$ 3 | 47 (26.9) | 128 (73.1) | 1.74 (1.04–2.90) | | 0.66 (0.32–1.37) | |
| ART Regimen | | | | | | |
| TDF-based | 27 (20.5) | 105 (79.5) | 0.86 (0.51–1.45) | 0.566 | | |
| Non-TDF based | 51 (23.1) | 170 (76.9) | 1 | | | |
| Hemoglobin (g/dl) | | | | | | |
| $<$ 12 | 30 (42.9) | 40 (57.1) | 3.67 (2.09–6.47) | $<$ 0.001 | 3.38 (2.00–7.46) | 0.001 |
| $\geq$ 12 | 48 (17.0) | 235 (83.0) | 1 | | 1 | |
| Baseline eGFR (ml/min/1.73m$^2$) | | | | | | |
| $<$ 60 | 28 (40.0) | 42 (60.0) | 3.11 (1.76–5.48) | $<$ 0.001 | 1.10 (0.54–2.24) | 0.596 |
| $\geq$ 60 | 50 (17.7) | 233 (82.3) | 1 | | 1 | |

ART, antiretroviral therapy; eGFR, estimated glomerular filtration rate; TDF, tenofovir

underlying pathophysiologic mechanisms for the relationship between lipid levels and loss of renal function are not yet fully understood, although there are data that oxidative stress and insulin resistance may mediate the lipid-induced renal impairment [36].

About 22.1% of patients on ART in our study had renal impairment (eGFR $<$ 60 ml/min/ 1.73 m$^2$). This finding is higher than that reported from a study conducted in Northwest Ethiopia where renal impairment was found in 11.7% of the HIV-infected patients on ART [22], and from studies in Tanzania (1.2%) [21], and Ghana (5.2% and 9.9%) [37,38] who used the

same MDRD formula. The differences could be due to variation in the severity of HIV infection, types and duration of ART, or due to variation in the lifestyle and age distribution of study patients. Analysis of the present study showed a significant increase in the prevalence of renal impairment after a median ART duration of 3.0 years, from 19.8% at baseline to 22.1%. This was consistent with findings reported from the Ethiopian study; which showed an increase in prevalence of renal impairment from 3.6% before to 11.7% after ART [22]. In contrast, the above sub-Saharan African cohort studies reported a significant decrease in the prevalence of renal impairment after initiating ART [20,21]. These studies however, reported less renal impairment following ART initiation among individuals starting therapy with advanced disease, median CD4 cell counts $< 143$ cells/mm$^2$. The disparity can also be explained by differences in the etiologies of renal impairment. For example, HIV-related renal impairment, which responds to ART, is absent among Ethiopian descents [39,40] and ART is not always beneficial in improving renal function in this context [10,18].

The present study also revealed a decline in mean eGFR after ART initiation, from a mean eGFR of $112.9 \pm 81.2$ ml/min/1.73 m$^2$ at baseline to $93.9 \pm 60.6$ ml/min/1.73 m$^2$ after a median ART follow-up of 3.0 years. A study in Ghana similarly showed a progressive decline in mean eGFR over the period whilst on ART [37]. In the ICONA Foundation cohort study, Tordato et al. [35] observed a decrease in eGFR of at least 20% from pre-ART levels in patients on ART who were drug-naïve at baseline. In the Asian study by Suzuki et al [41], a faster rate of eGFR decline was also observed after ART initiation, with a median duration of 3.07 years. The decline in eGFR emerges in patients on ART for no less than 12 weeks after the start of combination ART, and then tended to decrease gradually thereafter [42]. Exposure to HIV viremia, chronic inflammation, and potentially nephrotoxic antiretroviral drugs, as well as structural or functional disorders in the context of ageing and associated comorbidities may play a role in the pathogenesis of eGFR decline in HIV-infected patients on ART [23,43]. However, an improvement in eGFR after the commencement of ART has been demonstrated in the Tanzanian, South African and Ugandaian cohort studies [20,21,32]. Our results also showed that among patients with baseline normal renal function (eGFR $\geq$90 ml/min/1.73 m$^2$), 49.7% experienced some degree of renal impairment, which is similar to recent studies done in Tanzania and China [29,33]. Likewise, in the analysis of patients who initiated ART with normal renal function a significant proportion of patients have been shown to experience a decrease in eGFR to a value of $< 60$ or $< 90$ mL/min/1.73 m$^2$ after ART initiation [35,44]. These findings have particular importance in our settings, where monitoring of renal function is often insufficient due to limited resources.

Older age, being female, low current CD4 count, and low hemoglobin were associated with renal impairment among HIV-infected patients on ART. These findings are similar to those of other related studies [29,32,33,35,37,44,45]. Older patients are at increased risk for renal impairment as a result of increasing comorbid risk factors such as diabetes and hypertension, which are prevalent in HIV-infected patients on ART, or due to a physiologic decline in GFR with age [13]. Aging and gender are a non-modifiable risk factors that played a significant role in the development of renal impairment in our patients during follow-up, suggesting the need to implement possible preventive measures in these sub-group of population for reducing progression to renal failure and other adverse events. On the other hand, maintenance of high CD4 counts and of normal hemoglobin levels may be potentially modifiable factors that can reduce the risk of renal impairment in HIV-infected patients on ART. Similar to the South African study, a high BMI has been identified as a risk factor for renal impairment in our patients after ART initiation [32]. High BMI after ART-initiation is associated with cardiometabolic complications including atherothrombotic cardiovascular disease and diabetes that increase the risk for renal impairment during HIV treatment [46,47].

Lower baseline CD4 count has also been identified as risk factors for renal impairment after ART initiation. This is consistent with findings from related studies, which reported that lower baseline CD4 count may have an additive effect on renal impairment risk in HIV infected patients after ART initiation [21,29,41,45]. We found no significant differences in renal impairment among patients on TDF and non–TDF containing ART regimens. Similar findings were observed in an African studies [21,32,48]. The Myanmar study by Kyaw et al examining nearly 1400 HIV infected patients starting first line TDF based ART has also shown a low incidence of renal impairment and suggested that routine assessment of renal function may not be needed among those on a TDF-containing regimens [45]. These findings are reassuring for clinicians in low income settings where TDF is massively prescribed as part of first-line ART regimen. However, statistically significant reductions in renal function after the initiation of TDF, compared with after the initiation of other drugs, have been reported in other studies [35,41,42]. In this regard, results from a systematic review identified studies in Africans reporting statistically significant renal function decline associated with TDF use [49]. In the Ethiopian cohort study by Yazie et al [50], an eGFR decrease of >25% from baseline was observed in 25.4% of patients initiating TDF containing combination ART. Differences in study design, definition of renal impairment and duration of TDF exposure may partially explain the discrepancy between the results of the present study and the previous study in the country. Future prospective study is therefore required with large sample size and long duration to ascertain the association of TDF with renal impairment among Ethiopian patients initiating TDF-based ART regimens.

## Limitations

Study limitations include that this is an observational cohort study and not a randomized clinical trial. We only included patients with measured creatinine values at baseline and during follow-up at their last visit. This might have introduced a bias of selection if having a creatinine was non-random, and patients included and excluded were different. However, from side of the clinic there was neither a selective process preferring specific patients for creatinine measure, nor did the analysis yield a difference between patients excluded from those included in the analysis with regard to age, sex and clinical parameters. Data of other contributing factors to renal impairment such as opportunistic infections and their related treatments, some clinical indicators of renal health or treatment (e.g. proteinuria and viral loads), as well as attributable causes of death were not available and thus could not be further investigated. Although subjects in this analysis were exposed to multiple ART regimens, the effects of specific ART regimens on renal function was not assessed. The MDRD study equation has not been modified for use in HIV-positive Ethiopian patients. Lastly, the measurement of serum creatinine was not standardized; this might influence the performance of eGFR equation. This also hindered us from using the popular Chronic Kidney Disease Epidemiology Collaboration equation.

## Conclusions

In conclusion, impaired renal function was common in HIV-infected patients initiating ART in an outpatient setting in Ethiopia, and there appears to be a high prevalence of renal impairment after a median ART follow-up of 3 years. Our data highlight the need for assessment of renal function at baseline before ART initiation and regular monitoring of renal function for patients with HIV during follow-up, with aggressive management of risk factors. Further well-controlled cohort studies for the evaluation of long-term effects of ART on renal function are needed to confirm our findings.

## Supporting information

**S1 Dataset. The Excel database used for this manuscript.**
(XLSX)

**S1 Table. Baseline characteristics of patients included and excluded from the analysis.**
(DOCX)

## Acknowledgments

The authors would like to acknowledge staffs of ART clinic of Mehal Meda Hospital for collecting the data.

## Author Contributions

**Conceptualization:** Temesgen Fiseha, Angesom Gebreweld.

**Data curation:** Temesgen Fiseha, Angesom Gebreweld.

**Formal analysis:** Temesgen Fiseha.

**Investigation:** Temesgen Fiseha, Angesom Gebreweld.

**Methodology:** Temesgen Fiseha, Angesom Gebreweld.

**Software:** Temesgen Fiseha, Angesom Gebreweld.

**Supervision:** Temesgen Fiseha, Angesom Gebreweld.

**Validation:** Temesgen Fiseha, Angesom Gebreweld.

**Visualization:** Temesgen Fiseha, Angesom Gebreweld.

**Writing – original draft:** Temesgen Fiseha.

**Writing – review & editing:** Temesgen Fiseha, Angesom Gebreweld.

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
