## [Decision Letter · Decision Letter 0]

25 Sep 2020

PONE-D-20-19005

Renal function in a cohort of HIV-infected patients initiating antiretroviral therapy in an outpatient setting in Ethiopia

PLOS ONE

Dear Temesgen Fiseha,

Thank you for submitting your manuscript to PLOS ONE. After careful consideration, we feel that it has merit but does not fully meet PLOS ONE’s publication criteria as it currently stands. Therefore, we invite you to submit a revised version of the manuscript that addresses the points raised during the review process.

We look forward to receiving your revised manuscript.

Kind regards,

Professor Kwasi Torpey, MD PhD MPH

Academic Editor

PLOS ONE

Journal Requirements:

2. Thank you for stating in the text of your manuscript " The study protocol was approved by the Institutional Review Board of College of Medicine and Health Sciences, Wollo University. "

* Please also add this information to your ethics statement in the online submission form.

3. In your ethics statement in the Methods section and in the online submission form, please provide additional information about the data used in your study. Specifically, please ensure that you have discussed whether all data were fully anonymized before you accessed them and/or whether the IRB or ethics committee waived the requirement for informed consent. If patients provided informed written consent to have data from their medical records used in research, please include this information.

Reviewers' comments:

Reviewer's Responses to Questions

**Comments to the Author**

1. Is the manuscript technically sound, and do the data support the conclusions?

Reviewer #1: Partly

Reviewer #2: Yes

Reviewer #3: Partly

2. Has the statistical analysis been performed appropriately and rigorously? 

Reviewer #1: No

Reviewer #2: Yes

Reviewer #3: Yes

3. Have the authors made all data underlying the findings in their manuscript fully available?

Reviewer #1: No

Reviewer #2: Yes

Reviewer #3: Yes

4. Is the manuscript presented in an intelligible fashion and written in standard English?

Reviewer #1: Yes

Reviewer #2: Yes

Reviewer #3: Yes

5. Review Comments to the Author

Reviewer #1: Review comments

Important study looking at the burden of Renal impairment among HIV infected adults in the ART era. However, there are issues that must be addressed first before first interpreting the findings and then considering this paper for publication.

Major Issues

1. First, patients with missing data were dropped and only those who had data were analyzed. If were are to infer the findings presented here we need to know that they represent the underlying population (denominator) and hence they are from a random sample of the target population. Until details are provided this data as it is suggests a high possibility of non-random selection of participants making the study prone to selection bias. To evaluate this study, we need to know number, proportion and characteristics of patients who were dropped. Usually a flow diagram will be useful to demonstrate this.

2. Secondly, the analytical approach used is incongruent with the study design. While a cohort design was used only the prevalence analyses and logistic regression where used. Actually for the baseline prevalence assessment was okay however upon follow-up a longitudinal analysis that accounts for time in follow-up should be used both for disease burden and multivariate regression. This means a time-to-event analysis and a pooled logistic regression or poisson regression or proportional hazards regression etc… should be considered. This warrants a reanalysis if this paper is to be considered for publication.

3. Since the outcome, renal impairment, could be precluded by death as a competing risk, this also will have to be considered as well with the appropriate survival analysis technique for the descriptive analysis and hence consider the Aalen-Johansen estimator.

4. Characteristics of the cohort are not described well. While time zero is stated as ART initiation, end of follow-up, transfer out, loss to follow up, death as a competing risk, database closure etc… are all not defined nor described. Also the respective numbers and proportions of how these outcomes affected the study population during follow-up are not provided. This must be addressed if we are to interpret the findings appropriately.

5. In this HIV infected population on ART, regressions didn’t consider any HIV related immological and virological parameters. Opportunistic infections that could impair the kidneys either directly or due to their treatments e.g. CCM and sepsis from any cause were also not accounted for either as mediators or confounders.

6. Another aspect that might need to be considered for the analysis is the within and between subject variation overtime for the renal impairment. Risk of renal impairment depends of the individual and varies with time depending on what the patients is exposed to.

Reviewer #2: The authors looked at the renal function in a cohort of HIV patients in Ethiopia. The manuscript is well written and the language is acceptable. My comments are attached as sticky pads in the attached PDF file.

Reviewer #3: This is a pertinent topic which the authors can improve.

1. The introduction would benefit from a better detailed literature review that puts into better context the problem statement for Ethiopians . A more robust literature review is needed to strengthen the article.

2. The study method, while it is a retrospective cohort, should be better stated as Retrospective "observational" cohort as this describes upfront the specific cohort methodology employed.

3. In line 114, authors state, "The inclusion criteria were age older than 18 years..." This implies that patients who were 18 years were excluded. The statement should be re-written to "...were 18 years and older" as it is clear they included 18 year olds in the study.

4. Lines 117-118. "The study protocol was approved by the Institutional Review Board of College of Medicine and Health Sciences, Wollo University." Authors must provide the approval number for the study.

5.The authors declare that this study did not obtain ethical approval. This is a study that involves human participants and therefore requires ethical approval as it involves access to and use of medical records. If approval was not obtained, authors must explain why. in the very least, a waiver of ethical approval letter must be obtained from the ethics review committee.

6. The authors declare that all data is available in a public domain. Is this with reference to the primary medical records used for the study? If this is the case, ethics approval/consent from the patients or waiver thereof must be in place to provide for this.

7. May the authors also clarify where (the repository) the data in comment 6 is available?

8. Limitations; In stating the limitations to the study, line 306, outline how the limitation impacted the results.

9. Minor correction: Please correct the typographical error on Reference number 19, line 384. The spelling of Naive was captured incorrectly in the reference manager

6. PLOS authors have the option to publish the peer review history of their article (what does this mean?). If published, this will include your full peer review and any attached files.

Reviewer #1: No

Reviewer #2: **Yes: **Christian Obirikorang

Reviewer #3: No

---

## [Author Response · Author response to Decision Letter 0]

22 Oct 2020

Response to Reviewer Comments 

Reviewer #1 

Comment # 1: First, patients with missing data were dropped and only those who had data were analyzed. If were are to infer the findings presented here we need to know that they represent the underlying population (denominator) and hence they are from a random sample of the target population. Until details are provided this data as it is suggests a high possibility of non-random selection of participants making the study prone to selection bias. To evaluate this study, we need to know number, proportion and characteristics of patients who were dropped. Usually a flow diagram will be useful to demonstrate this.

Response #1: As suggested, it is stated in the result section as “Six hundred and forty-eight adults ≥ 18 years old were followed in this outpatient ART clinic for at least six months. A total of 295 were excluded: 143 did not have recorded baseline creatinine result before ART initiation, 40 did not have creatinine results after initiating ART at the most recent visit, and 112 patients had missing data or lost to follow up (Fig. 1). The excluded patients were similar to those who were included in this analysis in terms of age (mean 37.3 ± 10.0 years) and gender (60.0% female) distributions. The mean baseline CD4 count was 398.6 � 217 cells/mm3 in the excluded HIV-infected patients and 50 (16.9%) were in WHO clinical stage 3/4.” (Line 153-159)

Fig 1. Flow diagram showing the number of patients included in the analysis 

Comment # 2: Secondly, the analytical approach used is incongruent with the study design. While a cohort design was used only the prevalence analyses and logistic regression where used. Actually for the baseline prevalence assessment was okay however upon follow-up a longitudinal analysis that accounts for time in follow-up should be used both for disease burden and multivariate regression. This means a time-to-event analysis and a pooled logistic regression or poisson regression or proportional hazards regression etc… should be considered. This warrants a reanalysis if this paper is to be considered for publication.

Response #2: As stated in the introduction section the objective of our study was to evaluate the prevalence and associated risk factors of abnormal renal function before ART initiation (at baseline) and after ART initiation (after a median follow-up time of 3 years of ART treatment) and to describe the impact of ART on renal function in HIV-infected patients initiating ART using data collected before and after the ART treatment. (Line 102-109). Also as stated in the method section “After initiation of therapy, if a patient had multiple documented serum creatinine results, only the most recent result was utilized for the analysis” (Line 102-109), serum creatinine readings at the most recent measurement (i.e., at the most recent follow-up visit) was used for renal function assessment after ART treatment (during ART follow-up). As such statistical analysis of one group pre- and post-treatment comparison with one post-treatment measurement was employed, and it was stated in the method statistical analysis section as “Paired t-test was used to evaluate mean eGFR and CD4 T cell count difference before and after ART initiation. McNemar’s test was used to compare the proportion of patients with renal impairment before and after ART initiation.” (Line 144-146)

Comment # 3: Since the outcome, renal impairment, could be precluded by death as a competing risk, this also will have to be considered as well with the appropriate survival analysis technique for the descriptive analysis and hence consider the Aalen-Johansen estimator.

Response #3: As stated in response # 2, our objective was to evaluate the prevalence and associated risk factors of abnormal renal function before (at baseline) and during ART and to describe the impact of ART on renal function. HIV-positive patients are observed/measured twice and we compared values of outcome variable before and after ART treatment. Univariate and multivariate regression analysis were used to determine factors associated with renal impairment at the baseline prior to and after ART treatment. The present study does not focus on survival analysis/longitudinal analysis, because data on death at each time point; i.e., ≥ 3 time-to-event occurrence are needed (which is beyond our objective).

Comment # 4: Characteristics of the cohort are not described well. While time zero is stated as ART initiation, end of follow-up, transfer out, loss to follow up, death as a competing risk, database closure etc… are all not defined nor described. Also the respective numbers and proportions of how these outcomes affected the study population during follow-up are not provided. This must be addressed if we are to interpret the findings appropriately.

Response #4: Characteristics of the subjects described in the manuscript and in Table 1, and the proportion of patients excluded from the study are also described, as stated in response # 2. We collected repeated measures data where subjects are only measured twice, i.e., we have only one group of subjects and have measured them on two occasions only, before (at the baseline) and after ART initiating (at their last visit). As such the respective numbers and proportions of how these outcomes affected the study population during follow-up are not provided, a situation where more than two (≥ 3) measurements per subject are needed. 

Comment # 5: In this HIV infected population on ART, regressions didn’t consider any HIV related immological and virological parameters. Opportunistic infections that could impair the kidneys either directly or due to their treatments e.g. CCM and sepsis from any cause were also not accounted for either as mediators or confounders.

Response #5: HIV related immological parameter (CD4+ T cell count) was considered in the study analysis both before and after ART initiation and as stated in the result section (Line 179-180), paired t-test was used to assess mean differences for CD4+ T cell count before and after ART initiation. Being resource limited setting, measurements of virological parameters (viral load) are rarely performed in our setting and thus are not considered in the study analysis. Opportunistic infections that could impair the kidneys either directly or due to their treatments e.g. CCM and sepsis from any cause were also not accounted for either as mediators or confounders because such data are missing for most of the study patients, especially during ART at their last visit.

 Comment # 6: Another aspect that might need to be considered for the analysis is the within and between subject variation overtime for the renal impairment. Risk of renal impairment depends of the individual and varies with time depending on what the patients is exposed to.

Response #6: In the present study repeated measurements are taken from one group of participants on two occasions only, before (at baseline) and after ART treatment (at their last visits), differences in mean eGFR or differences in the proportion of patients with renal impairment before and after ART was compared using a paired sample test (within-subjects) as stated in response # 2. Compressions between groups both at the baseline and on ART were made using Chi-square test for categorical variables and student’s t-test for continues variables, as stated in the method statistical analysis section (Line 1141-142). As such between subject variation overtime (between-subject analysis) to compare renal function for two groups of participant (i.e., treatment and control groups) was not performed. Also analysis of repeated measures taken overtime (involving ≥3 measurements) were not performed using a linear regression models approach to assess the effect of time-dependent covariates on eGFR or to exhibit correlation within an individual overtime. 

Reviewer #2

The authors looked at the renal function in a cohort of HIV patients in Ethiopia. The manuscript is well written and the language is acceptable. My comments are attached as sticky pads in the attached PDF file.

Comment # 1: How were these patients factored in the data analysis since diabetes and hypertension are confounding factors for CKD?

Response #1: Diabetes and hypertensive patients had increased odds of renal impairment at baseline (Mocroft et al. 2007). These patients are factored in the data analysis to investigate these comorbid renal risk factors are associated with increased risk of prevalent renal impairment (non-HIV associated renal impairment), because they were thought to present significant challenges to HIV management especially in resource limited settings like our setting. Strategies that use baseline comorbidities, such as diabetes and hypertension, to identify individuals at high risk of renal impairment at the time of ART initiation may be effective.

Comment # 2: Aside comparing your study with other similar/different studies, what is accounting for the results obtained. What are the reasons/pathophysiology for the results obtained?

Response #2: As suggested, it is stated as “The disparity seems to be explained mainly by the characteristics of studied patients or by the differences in the creatinine assays and calibration. It can thus be inferred that HIV as a disease state is associated with renal function impairment independent of the possible nephrotoxic effects of antiretroviral agents used in the treatment of persons with HIV. The basis of HIV-induced renal impairment has been recognized as direct cellular injury by the virus or by changes in the release of cytokines during HIV infection (19,31).” (Line 239-244)

Comment # 3: There is a fine mechanism that explains the relationship between dyslipidamia (high cholesterol) and renal impairment.

Response #3: As suggested, it is stated as “The underlying pathophysiologic mechanisms for the relationship between lipid levels and loss of renal function are not yet fully understood, although there are data that oxidative stress and insulin resistance may mediate the lipid-induced renal impairment (36). (Line 253-256)

Comment # 4: What are the reasons accounting for the increase or decrease in the in eGFR in patients on ART?

Response #4: As suggested, it is stated as “Exposure to HIV viremia, chronic inflammation, and potentially nephrotoxic antiretroviral drugs, as well as structural or functional disorders in the context of ageing and associated comorbidities may play a role in the pathogenesis of eGFR decline in HIV-infected patients on ART (23,43).” (Line 281-284)

Comment # 5: Age is a confounding factor for a decline in eGFR. How different is that in HIV patients?

Response #5: It is different in HIV-infected patients, particularly in patients exposed to ART. As suggested, it is stated as “Older patients are at increased risk for renal impairment as a result of increasing comorbid risk factors such as diabetes and hypertension, which are prevalent in HIV-infected patients on ART, or due to a physiologic decline in GFR with age (13).” (Line 295-297)

Comment # 6: What accounts for a high BMI being a risk factor for renal impairment?

Response #6: As suggested, it is stated as “ High BMI after ART-initiation is associated with cardiometabolic complications including atherothrombotic cardiovascular disease and diabetes that increase the risk for renal impairment during HIV treatment (46,47).” (Line 304-307) 

Comment # 7: Reasons for these differences in 'similar' populations should have been stated.

Response #6: As suggested, it is stated as “Differences in study design, definition of renal impairment and duration of TDF exposure may partially explain the discrepancy between the results of the present study and the previous study in the country.” (Line 323-325)

Reviewer #3

This is a pertinent topic which the authors can improve.

Comment # 1: The introduction would benefit from a better detailed literature review that puts into better context the problem statement for Ethiopians. A more robust literature review is needed to strengthen the article.

Response #1: As suggested, it is stated as “Despite the introduction of the ‘test and treat’ program in Ethiopia, little is known about the status of renal function in HIV-infected individuals at the time of initiation of ART and during ART in this resource-limited settings. Renal function pre-ART and during ART is not routinely performed. The aim of … “(Line 105-108)

Comment # 2: The study method, while it is a retrospective cohort, should be better stated as Retrospective "observational" cohort as this describes upfront the specific cohort methodology employed.

Response #2: As suggested, it is stated as “Retrospective observational cohort study” (Line 32, 114)

Comment # 3: In line 114, authors state, "The inclusion criteria were age older than 18 years..." This implies that patients who were 18 years were excluded. The statement should be re-written to "...were 18 years and older" as it is clear they included 18 year olds in the study

Response #3: As suggested, it is stated as "...were 18 years and older" (Line 117)

Comment # 4: Lines 117-118. "The study protocol was approved by the Institutional Review Board of College of Medicine and Health Sciences, Wollo University." Authors must provide the approval number for the study.

Response #4: As suggested, it is stated as "..., Wollo University (# 133/13/12).” (Line 122)

Comment # 5: The authors declare that this study did not obtain ethical approval. This is a study that involves human participants and therefore requires ethical approval as it involves access to and use of medical records. If approval was not obtained, authors must explain why. in the very least, a waiver of ethical approval letter must be obtained from the ethics review committee.

Response #5: As suggested, it is stated as “Written informed consent was not given by participants for their clinical records to be used in this study, but patient’s identifiers were removed and only code numbers were used throughout the study.” (Line 122-124) 

Comment # 6: The authors declare that all data is available in a public domain. Is this with reference to the primary medical records used for the study? If this is the case, ethics approval/consent from the patients or waiver thereof must be in place to provide for this.

Response #6: This is not with reference to the primary medical records, but for the database used in this manuscript. As stated in response # 5; To ensure confidentiality, patient’s identifiers were removed and only code numbers were used throughout the study. 

Comment # 7: May the authors also clarify where (the repository) the data in comment 6 is available?

Response #7: The data used in the study manuscript will be available as a supplementary file. As suggested, it is stated as “The database for this manuscript is available as an additional supporting information (S1 Dataset).” (Line 124-125) Supporting Information. S1 Dataset. The Excel database used for this manuscript. (XLS) (Line 347-349)

Comment # 8: Limitations; In stating the limitations to the study, line 306, outline how the limitation impacted the results.

Response #8: This is to mean that the study lacked a control group or participants are not randomized to 2 arms (a randomized control-group pre- and post- treatment study is required to investigate the effects of ART on renal function).

Comment # 9: Minor correction: Please correct the typographical error on Reference number 19, line 384. The spelling of Naive was captured incorrectly in the reference manager

Response #9: As suggested, it is spelled as "Naive” (Line 407)

Looking forward to hearing from you. Thank you again for your consideration! 

Sincerely, 

Temesgen Fiseha (BSc, MSc)

---

## [Decision Letter · Decision Letter 1]

12 Nov 2020

PONE-D-20-19005R1

Renal function in a cohort of HIV-infected patients initiating antiretroviral therapy in an outpatient setting in Ethiopia

PLOS ONE

Dear Dr. Fiseha,

Thank you for submitting your manuscript to PLOS ONE. After careful consideration, we feel that it has merit but does not fully meet PLOS ONE’s publication criteria as it currently stands. Therefore, we invite you to submit a revised version of the manuscript that addresses the points raised during the review process.

We look forward to receiving your revised manuscript.

Kind regards,

Kwasi Torpey, MD PhD MPH

Academic Editor

PLOS ONE

Additional Editor Comments (if provided):

All comments addressed except for Reviewer 1. This needs to be addressed in the revision

Reviewers' comments:

Reviewer's Responses to Questions

**Comments to the Author**

1. If the authors have adequately addressed your comments raised in a previous round of review and you feel that this manuscript is now acceptable for publication, you may indicate that here to bypass the “Comments to the Author” section, enter your conflict of interest statement in the “Confidential to Editor” section, and submit your "Accept" recommendation.

Reviewer #1: (No Response)

Reviewer #2: All comments have been addressed

Reviewer #3: All comments have been addressed

2. Is the manuscript technically sound, and do the data support the conclusions?

Reviewer #1: No

Reviewer #2: Yes

Reviewer #3: Yes

3. Has the statistical analysis been performed appropriately and rigorously? 

Reviewer #1: No

Reviewer #2: Yes

Reviewer #3: Yes

4. Have the authors made all data underlying the findings in their manuscript fully available?

Reviewer #1: Yes

Reviewer #2: Yes

Reviewer #3: Yes

5. Is the manuscript presented in an intelligible fashion and written in standard English?

Reviewer #1: Yes

Reviewer #2: Yes

Reviewer #3: Yes

6. Review Comments to the Author

Reviewer #1: Review comments PONE D2019005R1

I thank the authors for attempting to resolve the outstanding issues with this important work. However major challenges remain mostly a valid reanalysis of the work for it to provide useful information to current practice/ care in the region.

1. Findings based on this study are for a finite unrepresentative highly selected population to which inferences will be made given the nature of population used in this study. There is a high likelihood of unequal selection based on exposures/predictors and on whether one had a creatinine result or was alive or not lost to follow-up. It is stated that those excluded were similar to those included based on Age and sex etc, but nothing is stated in terms of WHO stage i.e. all clinical parameters so that we are confident that the presented findings could be representative of the target population. Also since this is a cohort being followed for change in renal function, we cannot divorce ourselves from the challenges related to cohort follow-up and these include: Loss to follow-up and competing risks in this case death prior to next creatinine. At a minimum these must be reported and how they were dealt with. Also the study population cannot be selected based on having or not having an outcome measurement otherwise we have unequal selection probability affecting participation hence selection bias. Also there is nothing in the paper to ascertain whether having or not having a creatinine was random or non-random.

2. A wrong analysis was used—This study addresses the question of what the impact of ART use is on renal function within the context of an observational study. The question is how does use of ART change renal function overtime? First creatinine is a continuous outcome that could have benefited from an analysis that looks at it in its native data form while also accounting for within and between subject variability i.e mixed effects linear regression or Generalized Estimating Equations (GEE). Dichotomizing/categorizing and then using paired T-Tests and McNemar tests are acceptable with clinical trials but with this non-experimental study design leaves results from this sort of analysis confounded and hence biased. Logistic regression was used in the multivariate analysis doesn’t account for the correlated/dependent nature of the creatinine outcome measurements. Options are GEE or Generalized Linear mixed models (GLMM) analyses.

3. There is unmeasured confounding--A common cause for ART regimen changes/use that could also affect renal function are various opportunistic infections as a results of clinical failure as well as there related treatments. These OIs are still not measured as stated by the authors. Also nothing about this bias is stated regarding their potential impact on the findings even in the limitations.

4. Residual confounding--Also while OIs most of which are WHO stage III or IV conditions are not measured these could partially have adjusted for with WHO stage in the regression models. The current models don’t account for this hence the current estimates have residual confounding. Can the authors address this in their analysis. This is one of the flaws of backward selection since it doesn't account for empirical inherent biologically plausible relationships in the data for face/ content validity of the models.

5. Reverse causality--Are the predictors being reported predictors of ART use on renal function or predictors of survival with poor renal function and ART use? With the current design while the outcome of interest is incident (new ) renal failure after ART start, the authors actually insist on looking at prevalence at two time points baseline and sometime after ART start i.e. disease burden at two time points hence the difference in prevalence being attributed to ART use. Also patients with prevalent renal function at baseline are followed to this second time point. In this sort of analysis since you have prevalent disease at baseline being followed to the next time point we are unclear whether predictors are for survival or are causal. In fact given that a patient has baseline and a repeat creatinine suggests that they might have had some clinical sine qua non for poor renal function i.e. How sure are we that these patients were not likely to have clinical symptoms that require follow-up for renal impairment hence findings suggesting a high and uncommon 50% prevalence after 3 months on ART.

Reviewer #2: (No Response)

Reviewer #3: My comments have been sufficiently addressed. Just a few minor typographical/grammatical errors I have now noted:

Line 75- please correct spelling of "loss"

Line 81-82: Please revise sentence beginning with "Renal function...." to include the appropriate verb.

7. PLOS authors have the option to publish the peer review history of their article (what does this mean?). If published, this will include your full peer review and any attached files.

Reviewer #1: No

Reviewer #2: No

Reviewer #3: No

---

## [Author Response · Author response to Decision Letter 1]

27 Dec 2020

Response to Reviewer Comments 

Reviewer #1 

Comment # 1: Findings based on this study are for a finite unrepresentative highly selected population to which inferences will be made given the nature of population used in this study. There is a high likelihood of unequal selection based on exposures/predictors and on whether one had a creatinine result or was alive or not lost to follow-up. It is stated that those excluded were similar to those included based on Age and sex etc, but nothing is stated in terms of WHO stage i.e. all clinical parameters so that we are confident that the presented findings could be representative of the target population. Also since this is a cohort being followed for change in renal function, we cannot divorce ourselves from the challenges related to cohort follow-up and these include: Loss to follow-up and competing risks in this case death prior to next creatinine. At a minimum these must be reported and how they were dealt with. Also the study population cannot be selected based on having or not having an outcome measurement otherwise we have unequal selection probability affecting participation hence selection bias. Also there is nothing in the paper to ascertain whether having or not having a creatinine was random or non-random.

Response #1: As suggested this is included as the limitation of the study and it is stated as We only included patients with measured creatinine values at baseline and during follow-up at their last visit. This might have introduced a bias of selection if having a creatinine was non-random, and patients included and excluded were different. However, from side of the clinic there was neither a selective process preferring specific patients for creatinine measure, nor did the analysis yield a difference between patients excluded from those included in the analysis with regard to age, sex and clinical parameters.” Line 331-336. Regarding the CD4 count and WHO clinical stage of the excluded patients, it is already stated as The mean baseline CD4 count was 398.6 ( 217 cells/mm3 in the excluded HIV-infected patients and 50 (16.9%) were in WHO clinical stage 3/4" Line 158-159. But, as suggested this information is provided as supporting information and it is stated as (See S1 Table, which shows characteristics of patients included and excluded from the analysis). Line 159-160. S1 Table. Baseline characteristics of patients included and excluded from the analysis. (DOCX) Line 361-362

Comment # 2: A wrong analysis was usedThis study addresses the question of what the impact of ART use is on renal function within the context of an observational study. The question is how does use of ART change renal function overtime? First creatinine is a continuous outcome that could have benefited from an analysis that looks at it in its native data form while also accounting for within and between subject variability i.e mixed effects linear regression or Generalized Estimating Equations (GEE). Dichotomizing/categorizing and then using paired T-Tests and McNemar tests are acceptable with clinical trials but with this non-experimental study design leaves results from this sort of analysis confounded and hence biased. Logistic regression was used in the multivariate analysis doesnt account for the correlated/dependent nature of the creatinine outcome measurements. Options are GEE or Generalized Linear mixed models (GLMM) analyses.

Response #2: In order to examine the effects of ART on change in renal function, difference in mean eGFR levels (as a continuous variable) among same group of people before and after antiretroviral treatment was compared using paired-sample t-test (analysis that looks at it in its native data form), and change in proportions (eGFR category) was compared using McNemars test. The paired t-test and McNemars test are the most appropriate tools for analyzing pre-post differences in observations using matched/paired data when same individuals are measured (or surveyed) twice, typically before and then after some kind of treatment/ intervention in order to assess its effectiveness. They are particularly applicable to testing the effectiveness of treatments with before and after designs such as matched and crossover clinical trials, matched cohort studies and matched case-control studies. (Refer Ross et al. Paired Samples T-Test. In: Basic and Advanced Statistical Tests; Biostatistics Open Learning Textbook [https://bolt.mph.ufl.edu/6050-6052/unit-4b/module-13/paired-t-test]; Pagano and Gauvreau, Principles of Biostatistics, 2nd edition; McNemar Q. Note on the sampling error of the difference between correlated proportions or percentages; McNemar's test using SPSS Statistics [https://statistics.laerd.com/spss-tutorials/mcnemars-test-using-spss-statistics.php]; Adedokun et al. Analysis of Paired Dichotomous Data: A Gentle Introduction to the McNemar Test in SPSS; Fagerland MW et al. The McNemar test for binary matched-pairs data: mid-p and asymptotic are better than exact conditional). Similarly, recent observational retrospective cohort studies by Akilimali et al. (doi: 10.1371/journal.pone.0140240), Johannessen et al. (doi:10.1186/1471-2334-11-190), Bekolo et al. (doi:10.1186/1471-2334-14-519), Adane et al. (PMID: 22519158), Peters et al. (doi:10.1038/ki.2008.305) used the paired sample t-test and McNemars test to assess the difference in response variable at two time points.

Comment # 3: There is unmeasured confounding--A common cause for ART regimen changes/use that could also affect renal function are various opportunistic infections as a results of clinical failure as well as there related treatments. These OIs are still not measured as stated by the authors. Also nothing about this bias is stated regarding their potential impact on the findings even in the limitations.

Response #3: As suggested this is included as the limitation of the study and it is stated as Data of other contributing factors to renal impairment such as opportunistic infections and their related treatments, some clinical indicators of renal health or treatment (e.g. proteinuria and viral loads), as well as attributable causes of death were not available and thus could not be further investigated. Line 336-339

Comment # 4: Residual confounding--Also while OIs most of which are WHO stage III or IV conditions are not measured these could partially have adjusted for with WHO stage in the regression models. The current models dont account for this hence the current estimates have residual confounding. Can the authors address this in their analysis. This is one of the flaws of backward selection since it doesn't account for empirical inherent biologically plausible relationships in the data for face/ content validity of the models.

Response #4: Variable selection should start with the univariate analysis of each variable and variables that show significance (P<0.25) in the univariate analysis should be included in the multivariate analysis (Hosmer et al, 2013. Applied logistic regression). As such, a univariate analysis was done to sort variables candidate for multivariable analyses having value less than 0.25. As already stated in the method section (Line 148-150), all variables that were found to be significant in univariate analysis (P < 0.25) were included in the multivariate logistic regression model to identify factors independently associated with renal impairment both at baseline and during ART. Regarding OIs, it is included as limitation of the study as in response # 3 above. 

Comment # 5: Reverse causality--Are the predictors being reported predictors of ART use on renal function or predictors of survival with poor renal function and ART use? With the current design while the outcome of interest is incident (new) renal failure after ART start, the authors actually insist on looking at prevalence at two time points baseline and sometime after ART start i.e. disease burden at two time points hence the difference in prevalence being attributed to ART use. Also patients with prevalent renal function at baseline are followed to this second time point. In this sort of analysis since you have prevalent disease at baseline being followed to the next time point we are unclear whether predictors are for survival or are causal. In fact given that a patient has baseline and a repeat creatinine suggests that they might have had some clinical sine qua non for poor renal function i.e. How sure are we that these patients were not likely to have clinical symptoms that require follow-up for renal impairment hence findings suggesting a high and uncommon 50% prevalence after 3 months on ART.

Response #5: The risk factors being reported in the present study are the factors associated with the risk of having renal impairment both before initiation of ART (at baseline) and during exposure to different ART regimens (after initiation of treatment at their last follow-up visits). With the current before (or pre)–after (or post) treatment data, we insist on looking at prevalence at two time points baseline and sometime after ART start i.e. disease burden at two time points hence the difference in prevalence before and after antiretroviral treatment (were actually just interested in whether some people will move from impaired eGFR to normal eGFR and others from normal eGFR to impaired eGFR just randomly). Using repeated measurement of the same group of people at two time points (i.e., using paired data), we performed statistical analysis of changes in a measure before and after antiretroviral treatment or to assess the effect of ART on renal function. The present study cannot answer the question whether renal impairment is causally involved or is a marker for mortality during the course of treatment, because such a study would require a different design.

Reviewer #3 

Comment # 1: Line 75- please correct spelling of "loss"

Response #1: As suggested, it is corrected and spelled as loss Line 101

Comment # 2: Line 81-82: Please revise sentence beginning with "Renal function...." to include the appropriate verb.

Response #2: As suggested, it is revised and stated as Renal function test prior to and during ART Line 107-108

---

## [Editor Report · Decision Letter 2]

2 Jan 2021

Renal function in a cohort of HIV-infected patients initiating antiretroviral therapy in an outpatient setting in Ethiopia

PONE-D-20-19005R2

Dear Dr. Fiseha,

We’re pleased to inform you that your manuscript has been judged scientifically suitable for publication and will be formally accepted for publication once it meets all outstanding technical requirements.

Kind regards,

Professor Kwasi Torpey, MD PhD MPH

Academic Editor

PLOS ONE
---

## [Editor Report · Acceptance letter]

7 Jan 2021

PONE-D-20-19005R2 

Renal function in a cohort of HIV-infected patients initiating antiretroviral therapy in an outpatient setting in Ethiopia 

Dear Dr. Fiseha:

I'm pleased to inform you that your manuscript has been deemed suitable for publication in PLOS ONE. Congratulations! Your manuscript is now with our production department. 

Kind regards, 

on behalf of

Professor Kwasi Torpey 

Academic Editor

PLOS ONE